# Specific and Non-Specific Biomarkers in Neuroendocrine Gastroenteropancreatic Tumors

**DOI:** 10.3390/cancers11081113

**Published:** 2019-08-04

**Authors:** Andrea Sansone, Rosa Lauretta, Sebastiano Vottari, Alfonsina Chiefari, Agnese Barnabei, Francesco Romanelli, Marialuisa Appetecchia

**Affiliations:** 1Section of Medical Pathophysiology, Food Science and Endocrinology, Dept. of Experimental Medicine, Sapienza University of Rome, 00165 Rome, Italy; 2Internal Medicine, Angioloni Hospital, San Piero in Bagno, 47026 Forlì-Cesena, Italy; 3Endocrinology Unit, Regina Elena National Cancer Institute IRCCS, 00144 Rome, Italy

**Keywords:** neuroendocrine tumors, biomarkers, gastroenteropancreatic tumors, prognostic markers, diagnosis

## Abstract

The diagnosis of neuroendocrine tumors (NETs) is a challenging task: Symptoms are rarely specific, and clinical manifestations are often evident only when metastases are already present. However, several bioactive substances secreted by NETs can be included for diagnostic, prognostic, and predictive purposes. Expression of these substances differs between different NETs according to the tumor hormone production. Gastroenteropancreatic (GEP) NETs originate from the diffuse neuroendocrine system of the gastrointestinal tract and pancreatic islets cells: These tumors may produce many non-specific and specific substances, such as chromogranin A, insulin, gastrin, glucagon, and serotonin, which shape the clinical manifestations of the NETs. To provide an up-to-date reference concerning the different biomarkers, as well as their main limitations, we reviewed and summarized existing literature.

## 1. Introduction

Neuroendocrine tumors (NETs) are a heterogeneous group of rare malignancies which, from a clinical point of view, represent a true challenge for clinicians at all stages of the disease. The term “neuroendocrine” adequately describes the cell features in these conditions: The presence of dense core granules, similar to those found in serotonergic neurons, is the reason for the “neuro” term, whereas “endocrine”, as expected, refers to the secretive properties of these tumors [1,2]. NETs have been reported in almost all organs, although gastroenteropancreatic (GEP) tumors are the most common neoplasms; these NETs originate from the diffuse neuroendocrine system of the gastrointestinal (GI) tract and from pancreatic islets. In fact, it is assumed that the primary site would belong to the GI tract in almost two cases out of three, with the second most common primary site being the lung [2,3,4]. GEP NETs most commonly occur in patients between 50 and 60 years of age and are often encountered incidentally, such as during diagnostic or surgical interventions on the abdomen [3]. Recent reports suggest that the overall incidence of NETs is progressively increasing over time, seemingly more than justified by increasing diagnosis rates [4]: Several environmental or genetic factors are being investigated as possible contributors to this phenomenon.

NETs are usually classified as either low-grade, generally indolent tumors, or high-grade aggressive carcinomas [5,6]. Low-grade tumors are more often discovered much later in their clinical course due to rarely occurring symptoms, whereas the more dramatic onset of high-grade tumors is more closely associated with poor clinical prognosis. Another classification is based on the patterns of peptides and amines secretion and defines NETs as either functioning or non-functioning [7]. These biomarkers can influence the clinical course of the disease and might be suggestive of a neoplasm developing in the near future; however, symptoms can often be nuanced, and careful clinical assessment is needed before undergoing potentially expensive and/or invasive tests. Even functioning NETs are often clinically silent for many years, and signs and symptoms often only become clinically manifest when metastases are already present; however, among the many bioactive substances secreted by NETs, some could be included in the diagnostic process and together with imaging, clinical features and biopsy allow for a reliable diagnosis. Additionally, the same substances might be useful for prognostic and predictive purposes, as markers of treatment failure, cancer progression, or recurrence [8]. However, biomarkers should never be considered “screening” tools, given the rarity of NETs (less than 1% of all diagnosed malignancies [3]) and the wide array of confounding factors which might impair measurements’ reliability. It is, therefore, of the utmost importance to understand whether signs and symptoms encountered during routine clinical examination could be suggestive of a NET, and thus investigate biomarkers accordingly to clinical suspicion.

Neuroendocrine markers might be divided in two “categories”, namely specific and non-specific markers, with the former being produced by functioning NETs and varying according to hormone production and the latter being produced by virtually all NETs [3,9,10]. These markers are biologically active, and are, therefore, able to influence the clinical manifestations of the underlying NET: This is clearly demonstrated by the presence of different clinical syndromes, such as carcinoid syndrome or Zollinger-Ellison syndrome, which originate from the hypersecretion of the same substances which are “used” as markers of the disease. Assessment of the different patterns of biomarkers expression is, therefore, extremely useful in all phases of the diagnostic and therapeutic process. Investigating a single biomarker is rarely suggested, as additional information can easily be obtained by a more thorough investigation. Biochemical confirmation of a NET should always be followed by imaging techniques aimed at assessing tumor size and relationship to adjacent structures.

## 2. Non-Specific Markers

### 2.1. Chromogranin A

Chromogranin A (CgA) is a glycoprotein secreted by neurons and neuroendocrine cells, like all other proteins belonging to the granin family [11]. All granins—including Chromogranin B and C, as well as secretogranin II, III and IV—are precursors of biologically active substances [11]; for example, CgA is a precursor of pancreastatin, catestatin, and vasostatins I and II. While all granins could be secreted by neuroendocrine tumors, CgA is the only one routinely used in clinical practice: The assay has a high sensitivity [9] and good specificity [10], and CgA is secreted by most NETs, including malignant ones [9,12,13,14]. A significant, positive relationship (r = 0.9858, *p* ˂ 0.0001) has been reported between serum and plasma CgA, suggesting that either measurement provides an adequate estimate of circulating CgA [15].

However, despite being a promising biomarker for NETs, CgA is not always reliable in clinical practice, and caution is suggested in its interpretation. A two to threefold increase in the upper normal range should be enough to raise suspicion, but several factors should be considered before coming to premature conclusions. CgA should always be measured in the fasting state, as food intake is likely to increase CgA levels, therefore, increasing the risk of false positives [16]. Several drugs, most notably proton pump inhibitors (PPI) and steroids, as well as several oncological and non-oncological conditions, increase serum CgA levels (Table 1). Increased levels of CgA could prove the presence of a neuroendocrine differentiation for some non-neuroendocrine carcinomas: This has been reported for several neoplasms, such as prostate cancer, small-cell lung cancer, breast cancer, colon-rectal cancer [17,18]. Increased CgA levels have been detected in pancreatic adenocarcinoma and hepatocellular cancer as well: However, little is known concerning the potential pathophysiological meaning of the increase in CgA levels [16,19,20]. Elevated CgA levels have also been associated with a plethora of endocrine diseases, including neoplastic conditions, such as hyperthyroidism and hyperparathyroidism, as well as different endocrine tumors, such as pheochromocytoma, pituitary tumors, and medullary thyroid carcinoma. Additionally, CgA levels increase in non-neoplastic conditions, such as kidney failure, chronic atrophic gastritis, cardiovascular, and inflammatory diseases [16,19]. Treatment with PPIs induces hypergastrinemia, which in turn results in hyperplasia of enterochromaffin-like neuroendocrine cells: CgA levels can, therefore, increase (up to seven to tenfold) in patients undergoing therapy with PPIs, and elevated concentrations can be observed up to 2 weeks following treatment discontinuation [21]. Clinicians should, therefore, adequately investigate possible interfering factors before suggesting serum CgA measurement. Additionally, CgA is never a first-line diagnostic test and should not be considered a viable tool for screening [22] as the specificity of the CgA assay decreases up to 50% in populations with concomitant conditions, such as GI or renal disorders [23].

In addition to its role in tumor diagnosis, CgA is currently the most used liquid biomarker in the follow-up of NETs, as its concentration well correlates with disease progression and response to treatment [10] and a correlation between tumor burden and serum CgA has been proven. In fact, both advanced tumor stages and the presence of metastases correlate with serum CgA levels [24]; furthermore, a reduction in serum CgA concentrations in subjects undergoing treatment is a suggested surrogate marker of response to therapy. CgA levels decrease in cases of an adequate response, possibly even to the point of normalization, whereas persistently high concentrations are associated with poor clinical prognosis [25]. However, measurement of CgA is less reliable than advanced imaging techniques, such as MRI or CT, which can also provide the morphological information needed for RECIST (Response Evaluation Criteria In Solid Tumors) criteria [26] and which can, therefore, provide additional information concerning the outcomes of treatment. Different kits are available for measurement of circulating CgA: However, these assays might yield different results in relation to the forms of CgA and CgA-related peptide released from different neuroendocrine organs and NETs. A trial aimed at comparing three kits showed different sensitivity and specificity [27], suggesting that during follow-up, the same kit should always be used in all patients to prevent unreliable comparisons.

### 2.2. Neuron-Specific Enolase

Neuron-Specific Enolase (NSE) is a neuron-specific isomer of the enolase enzyme, found in neurons and neuroendocrine cells. Assessment of NSE alone is rarely adequate for diagnostic purposes of NETs, given that only 30 to 50%, in fact, secrete NSE [9,28,29]; additionally, NSE has low specificity and sensibility for differentiating NET from nonendocrine tumors [10]. Serum NSE concentrations are often increased in patients with other diseases, such as thyroid cancer, prostate carcinoma, neuroblastoma, and small cell lung carcinoma (SCLC) [8]; on the contrary, neuronal damage is associated with decreased levels of NSE [30]. In the presence of a pulmonary mass, high levels of NSE are often suggestive of an SCLC and have negative prognostic value. In fact, overexpression of NSE by all tumors, including NETs, is usually suggestive of poorly differentiated tumors, and thus of poor prognosis for higher-grade cancers [8]. Persistence of high NSE levels following treatment is usually considered a negative prognostic marker for SCLC, but there is no solid evidence in regard to post-treatment NSE levels for NETs. Assessing NSE and CgA at the same time as part of the diagnostic process could possibly increase the reliability of the measurement, providing further proof of the presence of a NET; however, given the non-specific nature of both markers, these tests provide little information concerning the localization of the primary tumor.

### 2.3. Pancreatic Polypeptide

Pancreatic polypeptide (PP) is mostly secreted by the pancreatic islet cells, whose function is largely unknown [31], although it is believed that regulation of pancreatic and GI secretion could be influenced by PP [9]. PP is generally considered a NET marker with low usefulness in clinical practice, due to its low sensitivity and specificity (63% and 81%, respectively) [32]; in fact, less than half of pancreatic NET patients show elevated serum PP [10]. Additionally, serum concentrations of PP can be increased by many factors, including physical exercise, hypoglycemia, and food intake [9], as well as decreased by somatostatin and hyperglycemia. A significant improvement in sensitivity can be obtained by the combination of PP with another marker, most commonly CgA [33,34]. As 93% of its secretion can be traced back to the F cells in the pancreas [31], PP is most likely suggestive for pancreatic NETs, and decline in PP levels following treatment is a good prognostic marker; however, increased serum concentration of PP have been reported for other forms of GI NETs as well [10], and as such a cautionary approach is always suggested.

### 2.4. Human Chorionic Gonadotropin

Human chorionic gonadotropin (hCG) is a heterodimeric glycoprotein physiologically synthesized during pregnancy by the placenta. As a protein heterodimer, hCG is composed of two different subunits, named α and β: The α subunit is shared with the pituitary hormones LH (luteinizing hormone), FSH (follicle-stimulating hormone) and TSH (thyroid stimulating hormone), whereas the β subunit (β-hCG) is unique. Different patterns of expression for hCG have long been identified in different endocrine as well as non-endocrine tumors [35], as tumors often lack the mechanisms to pair the two subunits: Increased expression of α subunit is common for pituitary tumors and NET, whereas pancreatic tumors often show increased secretion of β subunit. However, hCG is rarely used in everyday clinical practice for NETs [22]. In testicular tumors, combining hCG with other similar markers, such as alpha-fetoprotein (AFP), could improve the efficacy of the measurement [36]; however, assessment of hCG and AFP is generally not recommended in NETs, since both lack the sensitivity of specificity of CgA. In testicular cancers, however, hCG is also valuable as a prognostic marker.

### 2.5. Alpha-Fetoprotein

Alpha-fetoprotein (AFP) is a peptide hormone produced by the yolk sac and the fetal liver during development. In adults, AFP is primarily recognized as a biomarker for hepatocellular carcinoma [37] and testicular non-seminomatous germ cell cancer [38]. An increase in serum AFP has been reported in NETs, suggesting its possible role as a marker for diagnosis [39,40]; however, more recent evidence suggests that AFP might be a marker of cellular de-differentiation rather than a biomarker per se [36]. Decrease of AFP is often a sign of adequate treatment, although the validity of this finding in the context of NETs is not definite.

## 3. Specific Markers

### 3.1. Serotonin

Serotonin (5-hydroxytryptamine, 5-HT) is a biogenic peptide mainly secreted by the enterochromaffin cells of the small intestine [10], as well as by the serotoninergic neurons of the central nervous system [9]; additionally, 5-HT is also stored in the platelets. Two enzymes, tryptophan hydroxylase (TPH) and aromatic amino acid decarboxylase (DDC), are involved in the metabolic pathways which allow biosynthesis of 5-HT from its precursor, tryptophan. The biological functions of 5-HT include vasoconstriction, regulation of sleep, mood, appetite, and GI motility: The carcinoid syndrome resulting from excessive 5-HT is, therefore, characterized by flushing, diarrhea, wheezing, and dyspnea. The clinical features of carcinoid syndrome are strongly dependent on the concomitant secretion of other biologically active amines, such as tachykinins, prostaglandins, and kallikrein: The classic presentation resulting from isolated hyperproduction of 5-HT is the most common, occurring in almost 95% of cases [7,41,42,43]. Different symptoms, including bronchospasm, headache, and hypotension, are instead more common in the atypical carcinoid syndrome and are often related to histamine secretion, rather than 5-HT [7,44]. Almost 10 to 20% of patients with typical presentation develop carcinoid heart disease, also known as Hedinger’s syndrome, in which cardiac fibrosis and thickening of the heart valves leading to right heart failure [45]. This condition is possibly fatal and as such, should always be kept in consideration when performing a multi-disciplinary assessment of patients with carcinoid syndrome. Clinical suspicion of excessive 5-HT secretion needs biochemical confirmation: while several assays are available for 5-HT measurement, it is generally not recommended to perform such dosage, as fluctuations in secretion, as well as interindividual variations, make results unreliable. It is, therefore, recommended to assess urinary samples of 5-hydroxyindoleacetic acid (5-HIAA), the main metabolite of 5-HT, in order to obtain reliable information: Samples should be collected for up to 24 h, using plastic jars protected from direct light and pre-filled with acidic additive to keep pH below 3 (to ensure sterility and stability) [46]. The ENETS (European Neuroendocrine Tumor Society) Consensus Guidelines suggest measuring urinary 5-HIAA and serum CgA in all patients with NETs, both as part of the diagnostic process and during follow-up [47]. High-performance liquid chromatography (HPLC) is recommended, but automated assays or mass spectrometry could also be considered reliable methods for 5-HIAA measurement [48]. As for most biomarkers, several factors should be considered before drawing conclusions (Table 1): Despite good sensitivity and specificity (respectively, 70% and 90%), in fact, urinary 5-HIAA levels might be normal in non-metastatic patients and could also depend on tumor volume. Malabsorption, celiac disease, and tryptophan-rich foods [48,49] might lead to falsely elevated urinary 5-HIAA levels; similarly, renal failure or hemodialysis could result in falsely negative concentrations. Additionally, several drugs could influence urinary 5-HIAA levels, leading to both false positives and false negatives [48]; this is incredibly valuable during follow-up since changes in urinary concentrations prove useful as an objective marker of biochemical response to treatment with somatostatin analogs [9].

### 3.2. Gastrin

The G cells of the pyloric antrum, duodenum, and pancreas secrete gastrin, a peptide hormone whose main biological functions are promoting the release of gastric acid and stimulating GI tract motility [50]. Gastrin-secreting tumors—gastrinomas—are among the most common functioning NET affecting the pancreas, and are both sporadic and occurring as part of multiple endocrine neoplasms (MEN) type 1. Measurement of fasting serum gastrin (FSG) is suggested to suspect the presence of a gastrinoma and to diagnose Zollinger–Ellison syndrome (ZES): high FSG (often over ten times the upper limit of normal) and low gastric pH is required to perform diagnosis [9,51]. ZES prominently features gastroesophageal reflux and recurrent peptic ulcer disease, both resulting from excessive gastrin secretion; however, as gastrin increases in several conditions, most notably chronic atrophic fundus gastritis and the chronic use of PPIs (Table 1), it is generally not recommended to perform diagnosis based on FSG alone. To obtain a reliable measurement, PPIs should be interrupted 10 to 14 days before measuring FSG; to prevent peptic complications, high doses of H2 blockers (such as ranitidine 600 mg every 4 to 6 h) for 5 to 7 days could be used instead of PPIs. Provocative tests can be used to obtain more reliable results when FSG is only mildly increased, or when investigating patients undergoing treatment with PPIs [48]; It is worth remembering that treatment discontinuation is generally not recommended unless no mucosal damage is observed through gastroscopy, as persistent untreated hyperchloridria is associated with an increased risk of peptic complications [52]. Another approach involves using histamine type 2 (H_2_) blockers instead of PPIs for at least 1 to 2 weeks and then antacids for the last 2 days, before assessing FSG [53]. The most commonly used provocative test for the diagnosis of gastrinomas involves the use of secretin (2 U/kg body weight) administered by intravenous bolus: Gastrin serum is measured baseline (at −15 and −1 min before the test) and then after 2, 5, 10, 15, 20, and 30 min following secretin administration. An increase of ≥200 pg/mL at any time during the test is considered positive; however, using a delta of ≥120 pg/mL is considered an even more reliable method, with high sensitivity and specificity (94% and 100%, respectively). Additional tests have been reported, but the secretin test is usually preferred, being more sensitive and specific; however, these tests can be considered as further steps for confirmation in the case of an inconclusive secretin test. The calcium stimulation test is the most commonly used protocol in the presence of high level of clinical suspicion for ZES with a negative secretin test [54]: Serum gastrin levels are assessed every 30 min following the administration of calcium gluconate (5 mg/kg) over 3 h. An increase in serum gastrin >20% above baseline at 2, 4, or 6 min following i.v. calcium injection, usually with gastrin above 300 pg/mL, is usually conclusive for a positive result. The glucagon stimulation test is another stimulation test used for the diagnosis of gastrinomas. In this test, glucagon is infused at 20 µg/kg/h for 30 min: Diagnosis of gastrinoma is likely when the percentage of increase over the baseline gastrin peaks within 10 min following glucagon administration, with circulating gastrin over 200 pg/mL [55]. Additionally, the basal acid output (BAO) can support the diagnosis of ZES: A BAO >15 mmol/h is suggestive for this diagnosis [48]. The glucagon test can also be used post-operatively, as a measure of surgical efficacy: A negative response is a sign of adequate tumor removal and is associated with a decreased chance of recurrence [56]. Independently of the diagnostic test used for biochemical confirmation, localization of the tumor is mandatory to assess the chances for surgical treatment. The preferred imaging test is the Octreoscan (somatostatin receptor imaging using radiolabeled octreotide) together with CT; if needed, endoscopic ultrasound can also be used.

### 3.3. Insulin

Insulin is a polypeptide product of pancreatic islet β cells, mainly involved in energy balance and glucose metabolism and uptake. Insulin consists of two peptide chains, the A-chain and B-chain; β cells produce a precursor prohormone, proinsulin, which then undergoes cleaving by several endoproteases [57] before becoming mature insulin. The remaining, cleaved peptide sequence is called C peptide; though not directly involved in glucose metabolism, C peptide is a bioactive molecule, whose possible therapeutic uses have been hypothesized in the last decades [58]. Insulin-producing tumors—insulinomas—are almost exclusive to the pancreas and are the most common functioning neuroendocrine tumor for this organ [48]. Insulinomas are small neoplasms, therefore rarely associated with mass effect, differently from other NETs; almost 90% of insulinomas occur sporadically and are benign, with about 10% of cases occurring as part of MEN1 syndrome. Insulinomas show elevated serum insulin levels; however, other conditions might lead to increased serum insulin, most notably impaired glucose tolerance, diabetes mellitus type 2, administration of exogenous insulin (factitious hypoglycemia) and sulfonylurea-induced hypoglycemia [50]. The clinical features associated with hyperinsulinemia include both adrenergic and neuroglycopenic symptoms resulting from hypoglycemia; these symptoms often occur in the morning, although they can be present at any time during the day [48]. An insulinoma should be suspected when a patient has symptoms of hypoglycemia, serum glucose ≤40 mg/dL and improves following administration of glucose—a combination which has been defined as “Whipple’s triad” [59]. The gold standard for insulinoma diagnosis is the 72-h fasting test: In this cumbersome test, the patient is hospitalized and undergoes a blood sampling for serum glucose and insulin every 6 h, or whenever symptoms of hypoglycemia occur. The test is suspended when plasma glucose falls below the threshold of 55 mg/dL, and the patient develops classic symptoms of hypoglycemia; in 80% of the subjects, this requires 12 h, and almost 100% reach hypoglycemia by the end of the 72 h [48]. An insulinoma is diagnosed by the concomitant presence of hypoglycemia (≤40 mg/dL), inappropriately increased insulin levels (>6 U/L), and β-hydroxybutyrate levels ≤2.7 mmol/L. If test results of the 72-h fasting test are not conclusive despite clinical suspicion, it is generally recommended to perform a glucagon stimulation test immediately after: An increase in serum glucose levels following 1 mg glucagon administration is proof of adequate glycogen stores and can be observed in patients with insulinoma. Once a biochemical diagnosis has been established, imaging techniques are necessary for tumor identification: both CT and MRI can be used, although endoscopic ultrasound is the preferred technique.

### 3.4. Glucagon

Glucagon is a peptide hormone secreted by pancreatic α cells, with an opposite action compared to insulin as it stimulates glycogenolysis and gluconeogenesis. Elevated plasma glucagon levels, usually above 500 pg/mL, are generally observed only in glucagon-producing endocrine pancreatic tumors (glucagonomas) [8]; however, several conditions, such as cirrhosis, untreated diabetes mellitus, prolonged fasting, sepsis, burns, and acromegaly could increase serum glucagon levels. Glucagonomas are often large and malignant tumors, 20% occurring as part of MEN1 syndrome [60]: Clinical features of a glucagonoma syndrome include necrolytic migratory erythema, diabetes mellitus or impaired glucose tolerance, muscle wasting, and weight loss. The necrolytic migratory edema has a very distinctive appearance, with itchy rash on the perineum, thighs, and distal extremities prone to secondary infections. When hyperglucagonemia is associated with these symptoms, often in the presence of a liver metastatic disease with a large pancreatic mass, the diagnosis of glucagonoma is almost certain. A single glucagon measurement is rarely reliable, due to fluctuations in secretion; however, extremely elevated glucagon concentrations have been reported, even greater than 10,000 pg/mL, and are virtually diagnostic for a glucagonoma [60].

### 3.5. Somatostatin

Somatostatin is a peptide hormone whose synthesis mainly occurs centrally, in the hypothalamus, and peripherally, in the pancreatic δ cells, the gastric antral D cells and the APUD (Amine Precursor Uptake and Decarboxylation) cells [61]. Somatostatin is also called the growth hormone-inhibiting hormone, given its biological function in the regulation of the endocrine system; however, somatostatin also inhibits the release of the other pituitary hormones, namely TSH, ACTH (adrenocorticotropic hormone), and prolactin [62,63]. Additionally, somatostatin inhibits both glucagon and insulin secretion, decreases the rate of gastric emptying, and suppresses gastric acid secretion [63,64,65,66]. A somatostatin-secreting tumor, or somatostatinoma, should always be suspected in the presence of signs and symptoms associated with somatostatin excess: The most common clinical features are diabetes mellitus, diarrhea, and cholelithiasis, but weight loss and hypochlorhydria could also be associated with the elevated serum somatostatin levels. However, these symptoms are nonspecific, and somatostatinomas are often detected only much later in the progression of the disease, because of metastases with clinical manifestations [54]. Diagnosis of a somatostatinoma requires careful clinical and laboratory assessment: Somatostatin-secreting cells have been identified in a variety of extra-pancreatic NETs, including medullary carcinoma of the thyroid, pheochromocytoma, and paraganglioma.

### 3.6. Vasoactive Intestinal Peptide

Vasoactive intestinal peptide (VIP) is a peptide hormone which, as its name suggests, is mostly involved in vasodilation; however, VIP also increases glycogenolysis and stimulates gastrointestinal motility. VIP production and secretion both occur in several tissues, such as the gut and pancreas, hence, providing an explanation for its name, as well as in the brain, in the supra-chiasmatic hypothalamic nuclei. VIPomas—VIP secreting tumors—are rare tumors which most commonly develop in the pancreatic tail [67]. Despite a very typical clinical presentation (also defined as the Verner–Morrison syndrome) including watery diarrhea, electrolyte imbalance (mainly hypokalemia), and achlorhydria, these tumors have already metastasized at the time of diagnosis in more than 80% of the cases [60]. Laboratory assessment showing elevated serum VIP (>200 pg/dL) is often enough to diagnose VIPomas, but when symptoms of the Verner–Morrison syndrome are present, a threshold of 75 pg/mL can be adequate for suspicion.

## 4. Epi-GEP-NET-ics: Recent Developments in New Biomarkers

In recent years, the “omics” approach has allowed researchers to identify new biomarkers which might be correlated to clinical outcomes and which, potentially, might also be used in the clinical setting as better prognostic indicators. These include microRNAs (miRNAs), long non-coding RNAs (lncRNAs), circulating tumor cells, and DNA methylation patterns. So far, these biomarkers are prevalently used for research purpose in the field of neuroendocrine neoplasms, but a potential use as candidates for targeted therapies is not to be excluded.

### 4.1. miRNAs

MicroRNAs (miRNAs) are small, non-coding RNAs which control gene expression following transcription, either by inhibiting translation or by degrading specific mRNAs [68]; additionally, a potential endocrine-like role for miRNAs has been reported [69]. Dysregulation of miRNAs is considered to be a hallmark of human disease, including tumorigenesis; in the context of NETs, limited data have proven miRNA dysregulation in pancreatic and ileal NETs [70], and some miRNAs have been linked to either the primary tumor or to the presence of metastases. Both up and downregulation of several miRNAs are involved in the different features of neoplastic cells, such as the ability to maintain proliferative signaling while downregulating growth suppression or to invade adjacent structures (for a thorough review, see [71]). miRNAs are traditionally investigated in tissue samples, but they some studies have assessed the presence of circulating miRNAs, despite there being no identified source of secretion in the blood; in this regard, miRNAs also act as prognostic markers.

### 4.2. lncRNAs

Long non-coding RNAs (lncRNAs) are involved in pre-transcriptional regulation since they provide the necessary scaffolding for chromatin regulation [72]; however, a post-transcriptional role has been similarly suggested [73]. A single study by Modali et al. [74] has identified a potential role for the lncRNA MEG3 in insulinomas: MEG3 acts as a tumor suppressor in different cancers [75], and loss of function is associated with decreased control over cell proliferation.

### 4.3. Methylation

DNA methylation is perhaps the most known epigenetic alteration, occurring in cytosines preceding guanines—a dinucleotide generally addressed as CpG. Several CpG regions, or islands, are located at the 5′ end of the regulatory region of many genes. Such genes change their function according to their methylation status, with significant consequences on tumorigenesis: hypomethylation is associated with increasing chromosomal instability, whereas hypermethylation leads to decreased expression of tumor-suppressing genes [76]. Hypermethylation of several genes has been reported in regards to different NETs, such as RASSF1 [77], MGMT [78], TIMP-3 [79], and UCHL-1 [80]; given that each gene acts at a different level in regulation of cell cycle, the presence of several hypermethylated genes in the same subject is associated with increasingly advanced disease. Likewise, global hypomethylation has been more frequently observed in tumor samples from GEP-NETs than in healthy tissues, with different patterns for pancreatic and small bowel neoplasms [81,82].

## 5. Conclusions

In the clinical management of NETs, biochemical assessment of several peptides is an essential part of diagnosis and follow-up. Because early detection and treatment significantly improve the prognosis of NETs, clinicians should be well informed about their different clinical presentation (Table 2) as well as concerning the biochemical tests necessary for a correct diagnosis. However, measurement of these biologically active amines is not suggested for screening; generic tumor markers, such as CgA, can be increased in several clinical conditions, possibly leading to false positives. Provocative tests can be used for some markers, such as gastrin and insulin, to provide more reliable results, but it is generally not recommended to perform cumbersome testing in the absence of a clear indication. Clinicians should investigate these markers only when clinical features suggest doing so: If a NET is suspected, the choice of markers to assess should be guided by the presence of signs and symptoms, with imaging techniques providing additional information concerning tumor size and invasion of adjacent structures. Following treatment, the change in the production and release of these biomarkers could provide additional information in regards to the prognosis and response to therapy. While new diagnostic strategies are currently being developed, including the “-omics approach”, it is clear that “traditional” biomarkers of neuroendocrine differentiation still have a role in the approach to NETs.

## Figures and Tables

**Table 1 cancers-11-01113-t001:** Conditions, drugs, and foods interfering with serum biomarker assays.

	Biomarker	Increased by…	Reduced by…
**Non-specific markers**	Chromogranin A	Breast cancer, prostate cancer, ovarian cancer, hepatocarcinoma, pancreatic adenocarcinoma, colon cancer, kidney failure, heart failure, hyperthyroidism, hyperparathyroidism, chronic obstructive broncho–pulmonary disease, gastrointestinal pathologies, steroids, proton pump inhibitors	?
	Neuron-Specific Enolase	Thyroid cancer, prostate carcinoma, neuroblastoma, and small cell lung carcinoma	Neuronal damage
	Pancreatic Polypeptide	Physical exercise, hypoglycemia, food intake	Somatostatin, hyperglycemia
	Human Chorionic Gonadotropin	Pituitary tumors, pregnancy	?
	α-fetoprotein	Liver injury, pregnancy	?
**Specific markers**	Serotonin	Tryptophan-rich drugs (diazepam, ephedrine, phenobarbital, phentolamine…) and foods (peanuts, bananas, avocados, chocolate, vanilla, coffee, tea…), nicotine, malabsorption, celiac disease	Ethanol, ACTH, streptozocin, acetylsalicylic acid, heparin, MAO inhibitors, renal failure, hemodialysis
	Gastrin	Hypochlorhydria or achlorhydria, chronic atrophic gastritis, renal failure, *H. pylori* infection	Acetylsalicylic acid, levoDOPA
	Insulin	Hyperglycemia (including factitious and sulfonylurea-induced hypoglycemia), insulin resistance, insulinoma	Hypoglycemia, Type 1 Diabetes Mellitus, noradrenaline, fasting, glucagon
	Glucagon	Hypoglycemia, adrenaline, arginine	Hyperglycemia, somatostatin, insulin
	Somatostatin	?	?
	Vasoactive Intestinal Peptide	Bowel inflammation and ischemia	?

**Table 2 cancers-11-01113-t002:** Expression of different non-specific markers according to clinical manifestations. Features of tumors gastroenteropancreatic-neuroendocrine (GEP-NET)-associated clinical syndromes. 5-HIAA: 5-hydroxy-indolacetic acid; VIP: vasoactive intestinal peptide.

Syndrome	Symptoms	Biomarker
Carcinoid syndrome	Flushing Diarrhea Wheezing Dyspnea	Urinary 5-HIAA Serum 5-HIAA (less reliable)
Zollinger–Ellison syndrome	Recurrent peptic ulcer Gastro-esophageal reflux	Fasting serum gastrin Secretin stimulation test Calcium stimulation test Glucagon stimulation test
Insulinoma	Hypoglycemia	Insulin (72 h fasting) C-peptide Proinsulin Glucagon stimulation test
Glucagonoma	Necrolytic migratory erythema Diabetes mellitus or impaired glucose tolerance Muscle wasting Weight loss	Fasting serum glucagon
Somatostatinoma	Diabetes mellitus Diarrhea Cholelithiasis Weight loss Hypochlorhydria	Fasting serum somatostatin
VIPoma	Watery diarrhea Hypokalemia Achlorhydria	Serum VIP

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
