# Peer review of "Specific and Non-Specific Biomarkers in Neuroendocrine Gastroenteropancreatic Tumors"

_cancers, 2019, doi:10.3390/cancers11081113_

Round 1
Reviewer 1 Report
The article is very descriptive and lack the discussion in many parts. Some paragraphs describe guideline of using these biomarkers which are already published elsewhere for clinicians and they don’t add value in this article. Some paragraphs are totally taken from other reviews which indicates that this review failed in providing new insights for the readers. The article has heavily cited other review rather than the original articles (for instance the whole introduction section, L120, Table 1, line174-182, section 3.6).
Minor:
1- Title: the title should not include abbreviation (GEP).
2- The abstract didn’t state the aims of the review.
3- Line 61: the statement is unclear, using “both” is not appropriate
4- Line 132: this in controversy with the line 91.
5- Line 164: α-FP, why this biomarker has not been discussed in the article
6- Line 158: define LH & FSH
The article is very descriptive and lack the discussion in many parts. Some paragraphs describe guideline of using these biomarkers which are already published elsewhere for clinicians and they don’t add value in this article. Some paragraphs are totally taken from other reviews which indicates that this review failed in providing new insights for the readers. The article has heavily cited other review rather than the original articles (for instance the whole introduction section, L120, Table 1, line174-182, section 3.6).
Minor:
1- Title: the title should not include abbreviation (GEP).
2- The abstract didn’t state the aims of the review.
3- Line 61: the statement is unclear, using “both” is not appropriate
4- Line 132: this in controversy with the line 91.
5- Line 164: α-FP, why this biomarker has not been discussed in the article
6- Line 158: define LH & FSH
The article is very descriptive and lack the discussion in many parts. Some paragraphs describe guideline of using these biomarkers which are already published elsewhere for clinicians and they don’t add value in this article. Some paragraphs are totally taken from other reviews which indicates that this review failed in providing new insights for the readers. The article has heavily cited other review rather than the original articles (for instance the whole introduction section, L120, Table 1, line 174-182, section 3.6).
Minor:
1- Title: the title should not include abbreviation (GEP).
2- The abstract didn’t state the aims of the review.
3- Line 61: the statement is unclear, using “both” is not appropriate
4- Line 132: this in controversy with the line 91.
5- Line 164: α-FP, why this biomarker has not been discussed in the article
6- Line 158: define LH & FSH
Author Response
Dear Reviewer,
Thanks for your time and effort in reviewing our manuscript.
Below is a point-by-point response to your comments. We have tried to improve the manuscript according to your suggestions. We hope you might have positive review of the amended manuscript.
The article is very descriptive and lack the discussion in many parts. Some paragraphs describe guideline of using these biomarkers which are already published elsewhere for clinicians and they don’t add value in this article. Some paragraphs are totally taken from other reviews which indicates that this review failed in providing new insights for the readers. The article has heavily cited other review rather than the original articles (for instance the whole introduction section, L120, Table 1, line174-182, section 3.6).
The aim of our review was to provide an up-to-date, comprehensive review of the different biomarkers used for GEP-NETs, including their main limitations. We have summarised evidence from different papers, as well as reported summarised reports from other reviews, in order to have a manuscript which would fit our aim. We have cited additional papers in support of the evidence summarised by the cited reviews.
Minor:
1- Title: the title should not include abbreviation (GEP).
We have amended the title accordingly. Thanks.
2- The abstract didn’t state the aims of the review.
Thanks for pointing this out. We have stated the aims of our review.
3- Line 61: the statement is unclear, using “both” is not appropriate
We have revised the whole paragraph in order to reduce redundancy and improve the overall clarity of the sentence. Thanks for your comment.
4- Line 132: this in controversy with the line 91.
We have revised the sentence accordingly.
5- Line 164: α-FP, why this biomarker has not been discussed in the article
There is little evidence suggesting the possible role of AFP as a biomarker for NETs. However, we have decided to include a short section on AFP which, we believe, fits well in the context of our manuscript.
6- Line 158: define LH & FSH
Section 2.4 has been amended accordingly. Thanks.
Reviewer 2 Report
This is a timely and well-written review on the contribution of specific and non-specific biomarkers in neuroendocrine GEP tumors. The paper can be accepted pending the revision of the following points.
1. Is the affiliation of Dr. Romanelli the n.1? There is any n.6. Check it.
2. The authors should increase the legibility of the Table 1. Similar comments to Table 2.
3. I suggest that a facilitating version of the text should be introduced in two distinct tables describing the features of the non-specific and specific biomarkers as shown in the Table 1.
4. Discuss the contribution of genetic alterations in GEP-NETs tumorigenesis. In different papers it it has found methylation patterns, chromatin remodeling alterations, microRNA and long non-coding RNA (lncRNA) differential expression profiles that are peculiar of GEP-NETs, some of which are correlated with clinical outcomes. Several translational studies have provided robust results identifying potential prognostic biomarkers, and some of these have demonstrated preliminary success as serum biomarkers that can be used clinically.
5. It could be interesting to write a brief section on the current treatments of these tumors and if there are ongoing new trials.
Author Response
Dear reviewer, Thanks for your time and efforts in reviewing our manuscript. Here is a point-by-point response to all of your comments. We hope that the amended paper is now adequate for publication.
This is a timely and well-written review on the contribution of specific and non-specific biomarkers in neuroendocrine GEP tumors. The paper can be accepted pending the revision of the following points.
Thanks for your positive comment. We have tried to improve the manuscript following your suggestions.
Is the affiliation of Dr. Romanelli the n.1? There is any n.6. Check it.
Thanks for pointing this out. This was a minor mistake due to copy and paste for authors' affiliations.
The authors should increase the legibility of the Table 1. Similar comments to Table 2.
We have added a few thin grey lines which might help distinguishing the different biomarkers. Table 1 has also been re-built in order to make it easier to understand.
I suggest that a facilitating version of the text should be introduced in two distinct tables describing the features of the non-specific and specific biomarkers as shown in the Table 1.
We have expanded table 1 in order to provide a better overview of the different biomarkers routinely used in the clinical setting.
Discuss the contribution of genetic alterations in GEP-NETs tumorigenesis. In different papers it it has found methylation patterns, chromatin remodeling alterations, microRNA and long non-coding RNA (lncRNA) differential expression profiles that are peculiar of GEP-NETs, some of which are correlated with clinical outcomes. Several translational studies have provided robust results identifying potential prognostic biomarkers, and some of these have demonstrated preliminary success as serum biomarkers that can be used clinically.
We have written a new section (Section #4) detailing the new markers. However, we believe that providing a complete and thorough report of all epigenetic alterations described for GEP-NETs would be a topic worthy of a separate review, given the broad spectrum of new findings and their still uncertain relevance for clinical purposes.
It could be interesting to write a brief section on the current treatments of these tumors and if there are ongoing new trials.
Thanks for this interesting comment. Given that the review and the whole special issue are focused on biomarkers, we have included (wherever adequate evidence existed) an indication concerning changes in expression of the different biomarkers following treatment.
Round 2
Reviewer 1 Report
NA
This manuscript is a resubmission of an earlier submission. The following is a list of the peer review reports and author responses from that submission.